# The Association of Acute Phase Proteins in Stress and Inflammation-Induced T2D

**DOI:** 10.3390/cells11142163

**Published:** 2022-07-11

**Authors:** Tammy Speelman, Lieke Dale, Ann Louw, Nicolette J. D. Verhoog

**Affiliations:** Biochemistry Department, Stellenbosch University, Van der Byl Street, Stellenbosch 7200, South Africa; tammyspeel95@gmail.com (T.S.); liekedale@gmail.com (L.D.); al@sun.ac.za (A.L.)

**Keywords:** acute phase response, acute phase proteins, insulin resistance, type II diabetes, glucocorticoids, pro-inflammatory cytokines

## Abstract

Acute phase proteins (APPs), such as plasminogen activator inhibitor-1 (PAI-1), serum amyloid A (SAA), and C-reactive protein (CRP), are elevated in type-2 diabetes (T2D) and are routinely used as biomarkers for this disease. These APPs are regulated by the peripheral mediators of stress (i.e., endogenous glucocorticoids (GCs)) and inflammation (i.e., pro-inflammatory cytokines), with both implicated in the development of insulin resistance, the main risk factor for the development of T2D. In this review we propose that APPs, PAI-1, SAA, and CRP, could be the causative rather than only a correlative link between the physiological elements of risk (stress and inflammation) and the development of insulin resistance.

## 1. Introduction

Diabetes mellitus (DM) is one of the leading public health challenges worldwide. The global prevalence of diabetes is projected to increase from 537 million in 2021 to 783 million by 2045, a net increase of 46% [1]. In addition, it is among the ten leading causes of death worldwide [2]. Diabetes mellitus is classified as either: (i) gestational DM, (ii) type-1 DM (T1D) or (iii) type- 2 DM (T2D). The latter is the predominant form, comprising 90% of all DM cases. Therefore, a better understanding of T2D pathophysiology is of great importance. Although current treatments for T2D are often effective, they are linked to various side effects [3,4,5]. For example, metformin, a biguanide, commonly prescribed in patients diagnosed with T2D is linked to gastrointestinal side effects [6]. The usage of rosiglitazone, once widely prescribed to treat T2D, is currently restricted in most countries due to cardiovascular complications [6]. Therefore, novel therapeutic approaches are warranted.

T2D, a major non-communicable disease, is traditionally considered a metabolic disorder, which is mainly attributed to the initial development of insulin resistance [7,8]. The term ‘insulin resistance’ implies a reduced sensitivity of peripheral target tissues, which include adipose, muscle, and liver tissues, to normal circulating concentrations of insulin [9]. Although it is well established that insulin resistance is central to the pathogenesis of T2D [7,8], it remains unclear how this abnormality arises at a molecular level. Contrasting data exist on what the principal molecular perturbations are which lead to insulin resistance [10], although it does involve the insulin signaling pathway, an integrated network of signaling proteins and secondary messengers. A defect of or disruption to any of the signaling proteins or production of secondary messengers results in deficient insulin action, setting the scene for developing T2D [11,12]. 

Although numerous factors contribute to the development of T2D, including obesity, a common thread throughout the literature identifies inflammation and stress as key role players [13,14,15], with a close link between chronic inflammation and insulin resistance [16,17]. For this reason, T2D is regarded as a chronic, low-grade inflammatory state [18]. Inflammation is regulated by several biochemical mediators, of which cytokines are the most important. Pro-inflammatory cytokines such as tumor necrosis-alpha (TNF-α), interleukin-1 (IL-1), and interleukin-6 (IL-6), which are increased in response to obesity, induce insulin resistance at a molecular level by modulating the insulin signaling pathway [19,20,21,22,23,24,25,26,27]. Similarly, glucocorticoids (GCs), steroidal stress hormones, also cause insulin resistance in vivo [28,29]. Stress via GC signaling, like the above-mentioned pro-inflammatory cytokines, can trigger the acute phase response (APR), a part of the innate immune response, which has been reported to be activated in an insulin-resistant state (40).

TNF-α and IL-6, as well as GCs, induce the expression of several acute phase proteins (APPs), including plasminogen activator inhibitor-1 (PAI-1), serum amyloid A (SAA), and C-reactive protein (CRP) [30,31,32,33,34,35,36,37]. These APPs are routinely used as biological markers for T2D as their levels are significantly increased in the serum of T2D patients [38,39,40,41,42,43,44]. However, although associated with insulin resistance and thought to predict the development of T2D [45,46,47,48,49,50,51], whether these APPs could lead to the development of T2D remains to be elucidated. As an association exists between increased PAI-1, SAA, and CRP levels and the development of insulin resistance, it is possible that these APPs may be the causative link between inflammation and insulin resistance, however, evidence supporting this hypothesis is limited. In this review, the link between APPs and insulin resistance will be reviewed as a novel approach to understanding the development of GC- and inflammation-induced T2D. 

## 2. Insulin Resistance

The characteristic attenuated effect of insulin in peripheral tissues, indicative of insulin resistance, precedes the development of hyperglycemia [10,52,53]. Defective insulin action manifests itself as reduced glucose uptake in skeletal muscle and adipose tissue and increased glucose production in the liver, amongst other outcomes [54,55]. More specifically, insulin-induced glucose uptake via the glucose transporter type 4 (GLUT4) is restricted in both skeletal and adipocytes in an insulin-resistant state [56]. Additionally, glycogen synthesis in response to insulin is no longer promoted in the insulin-resistant liver and skeletal tissue and glycogenolysis is not suppressed [52]. This all leads to the inability of insulin to decrease blood glucose concentrations. In order to compensate for this effect, pancreatic β-cells increase the secretion of insulin, which results in hyperinsulinemia observed in insulin-resistant states and that is a primary contributor to the development of T2D [10,57,58], in addition to hyperglycemia [4]. Finally, when the β-cells, due to β-cell dysfunction, fail to produce the excess amounts of insulin needed, T2D emerges [59,60,61]. 

At the molecular level, two underlying mechanisms of insulin resistance have been proposed, both involving defective insulin signal transduction [52,62,63]. The first mechanism describes decreased activation of key nodes within the insulin signaling pathway, which include the insulin receptor (IR), insulin receptor substrate (IRS) proteins, and the central signaling protein, Akt [64,65,66,67,68]. For example, knockout of the IR as well as IRS proteins in rodent livers lead to hepatic insulin resistance, resulting in hyperglycemia and glucose intolerance [64,66,67]. Additionally, reduced tyrosine phosphorylation (and therefore reduced activation) of the IR and IRS proteins have been observed in insulin-resistant states [21,22,69,70], and hepatic inactivation of phosphoinositide 3-kinase (PI3K), phosphoinositide-dependent kinase-1 (PDK1), and mammalian target of rapamycin complex 2 (mTORC2). This results in the inactivation of Akt, which induces hyperglycemia and hyperinsulinemia in mice [71,72,73]. The second mechanism involves an imbalance between two pathways mediating insulin action: the PI3K/Akt pathway and the mitogen-activated protein kinase (MAPK) pathway. Under normal conditions, there is a balance between the PI3K/Akt pathway, responsible for the metabolic function of insulin and the mitogenic signaling by insulin controlled by components of the MAPK pathways, p38, ERK1/2, and JNK. However, dysregulation of insulin signal transduction shows an imbalance in this system [62,63,74,75]. Herein the PI3K/Akt pathway is inactivated, which disrupts nutrient homeostasis, while the activation of the MAPK pathway is sustained, promoting mitogenesis as well as increased serine/threonine phosphorylation (thus inactivation) of the IRS proteins, leading to the inhibition of the PI3K/Akt pathway [54,62,63]. This dysregulation can be caused by various factors, including the activation of inflammatory pathways, increased pro-inflammatory cytokines as well as stress and obesity [13,20,21,22,70,76,77,78,79,80,81,82,83,84].

Furthermore, insulin insensitivity in the different peripheral target tissues presents different phenotypes [53,85]: in the liver, hepatic glucose production is increased due to the inhibition of Akt-induced FoxO1 suppression as well as other transcription factors regulating glucose and lipid metabolism [52,54,63]. In adipose tissue, fat cell development is retarded and there is an increase in lipolysis [62,74]. The excessive free fatty acids travel to the liver and skeletal muscle, promoting gluconeogenesis and inhibiting glucose uptake, respectively, thus worsening hyperglycemia [62]. Additionally, hyperlipidemia, which is a key feature of insulin resistance, develops as a result of altered lipid metabolism, specifically in the liver, in which lipogenesis is increased [54,62]. Overall, insulin resistance is multifaceted and involves cross-talk between the peripheral target tissues [53,86,87,88,89,90] as well as various nodes within the insulin signaling pathway. 

Pickup and Crooke discussed how T2D may be considered a disease of the innate immune system [91]. The authors propose that T2D is an acute-phase disease, in which increased concentrations of pro-inflammatory cytokines and APPs are secreted, under the influence of various stimuli such as overnutrition [91,92]. In support, Rehman and Akash proposed that overnutrition is a major causative factor contributing to chronic inflammation [16]. APPs are evolutionary conserved proteins produced mainly in the liver in response to infection and inflammation [93] and their plasma levels have been associated with the complexities of T2D [38,91,92], leading to the question of whether they may play a more active role in development of the disease itself. 

## 3. Acute Phase Response (APR)

Homeostasis in mammals is ensured by several physiological mechanisms. When homeostasis is disturbed as a result of tissue injury, infection, and immunological disorders, the body responds by inducing a number of systemic and metabolic changes known as the APR [94,95]. 

The APR is a manifestation of the innate immune system [96] that comprises two reactions: local and systemic reactions [97]. The local reaction is initiated at the site of invasion or injury, which results in the release of pro-inflammatory cytokines, also known as early acute phase reactants [98]. These include IL-6, IL-1, and TNF-α, of which IL-6 is considered the main regulator of the APR in the liver [97,99]. The pro-inflammatory cytokines activate receptors on different target cells, which leads to intracellular signaling, resulting in the systemic reaction characterized by various physiological responses in different tissues. These include fever, leukocytosis, increased levels of GCs, activation of complement, changes in metabolism including increased gluconeogenesis, and finally synthesis of several plasma proteins, known as APPs [95,97,98,100]. The concentrations of APPs can either be increased (known as positive APPs) or decreased (known as negative APPs) in response to inflammatory stimuli [95,100]. Positive APPs are further classified into three categories, dependent on the magnitude of their response [101]. Upon stimulation, major APPs increase 10–1000-fold in concentration within 48 h followed by a rapid decline due to their short half-life [98,100,101]. In contrast, the increase in levels of moderate and minor APPs are much less pronounced, however, due to their longer half-life and, depending on the stimuli, have a longer duration (3–5 days) in circulation [98,100,101,102]. Thus, on average, the APR shows a rapid response that peaks within the first 48 h but can last up to 3–5 days. The biological functions of the different positive APPs are vast and involve activating the complement system (which also plays a role in T2D progression [103]), modulating the host’s immune response as well as wound healing and tissue repair [96,100]. 

Overall, the APR involving various APPs (each with a unique set of biological activities) is important to restore homeostasis [95] and lack of resolution of the inflammatory stimulus results in chronic inflammation [98]. A chronic APR has various disease implications: including T2D [104]. In fact, T2D is suggested to be an “acute phase disease” [91] and in support of this, numerous studies have reported increased levels of APPs; such as PAI-1, SAA, and CRP; in diabetes, [92,105,106]. Whether a chronic APR leads to the development of T2D is, however, unclear. It does, nevertheless, beg the question of whether these APPs play a role in the development of T2D during a sustained APR. 

## 4. Acute Phase Proteins

### 4.1. Plasminogen Activator Inhibitor-1 (PAI-1)

PAI-1, also named Serpin E1, belongs to a superfamily of serine-protease inhibitors (SERPINs). It is produced and released into circulation primarily by endothelial cells but also by other cell types, including hepatocytes and adipocytes [107]. The latter explains why PAI-1 is a well-known adipocytokine, as its levels are markedly increased along with the accumulation of fat [45,46,47,108]. This possibly explains the correlation between elevated PAI-1 levels and obesity, a risk factor for T2D. 

The main physiological role of PAI-1 is as key negative regulator of fibrinolysis through its role as the principal inhibitor of both urokinase- (u-PA) and tissue-plasminogen activator (t-PA) [109]. Under normal conditions, u-PA and t-PA are able to convert plasminogen to its active form, plasmin, which can degrade many blood plasma proteins, including fibrin clots in a process known as fibrinolysis. PAI-1 is therefore capable of inhibiting intravascular fibrinolysis, which leads to blood clotting or coagulation (hemostasis). Additionally, plasmin is able to degrade extracellular matrix (ECM) components, and therefore PAI-1 indirectly regulates ECM degradation [110], which is an important factor to consider when understanding the role of PAI-1 in different disease states. In addition to PAI-1’s role in hemostasis, it is thought to be involved in cell migration and remodeling of body tissues [107,111] 

Circulating PAI-1 levels vary more than any other component of the fibrinolytic system, possibly due to PAI-1 production being stimulated by a wide variety of signaling molecules, including IL-1, TNF-α, and insulin [112,113]. In addition, PAI-1 has been identified as a major stress-induced gene [114]. The activation of the hypothalamic-pituitary-adrenal axis by stressors, lead to an increase in the secretion of GCs, which are also able to induce PAI-1 expression [107,115]. In fact PAI-1 follows a similar circadian pattern as that of the endogenous GC, cortisol [110]. In healthy individuals, normal active PAI-1 plasma concentration ranges from 5–20 ng/mL [116]. This concentration range is suggested to be sufficient to control fibrinolysis [116]. However, under pathological conditions, several tissues produce substantial amounts of PAI-1 (15–36 ng/mL) in response to inflammatory cytokines. For example, elevated PAI-1 concentrations have been consistently observed in blood from T2D patients [39,40,117,118] to which hypofibrinolysis and atherothrombosis in individuals with T2D is attributed [110,111,119,120]. In addition, obese individuals, many of whom exhibit insulin resistance, were found to exhibit a three-fold elevation of PAI-1 in their blood, compared to lean individuals [121]. Elevated PAI-1 levels and hyperinsulinemia are also correlated [118,122]. The high expression levels of PAI-1 in these disease states raises the question of its contribution to the phenomenon. Indeed, PAI-1 was shown to be overexpressed in the adipose tissue of obese mice [123,124,125] and humans [117,126] and is considered a biological marker of obesity [127]. In obesity, PAI-1 affects adipocyte differentiation by inhibiting the degradation of ECM components (an important process during adipocyte differentiation) [110]. 

Clinically, improved control of hyperglycemia in patients with T2D decreases PAI-1 activity. Improving insulin resistance by diet, exercise, or oral antidiabetic drugs results in decreasing plasma PAI-1 levels [87,128]. For example, troglitazone, an antidiabetic drug was shown to decrease plasma PAI-1 antigen levels and activity in diabetic patients [129]. 

The Insulin Resistance Artherosclerosis study (IRAS) has found that the development of T2D could be predicted by high PAI-1 levels independently from other risk factors [39,40]. Whilst elevated PAI-1 levels are a core feature of obesity and insulin resistance, some studies have also linked PAI-1 to a direct causal role in these disease states (Table 1). Mice with PAI-1 deficiency, either through gene knockout or the use of a PAI-1 inhibitor, are protected from obesity including hyperglycemia and hyperinsulinemia and demonstrate improved insulin sensitivity [45,46,47,130,131]. Furthermore, PAI-1-deficient murine primary adipocytes exhibit enhanced insulin-stimulated glucose uptake and adipocyte differentiation is promoted [132]. In contrast, however, overexpression of PAI-1 in transgenic mice exhibited lower adipose tissue mass and total body weight [131,133] and PAI-1-deficient mice on a high-fat diet showed rapid adipose tissue development [134]. The differences observed in these mice studies could be attributed to the different genetic backgrounds of the mice as well as different protocols to induce obesity. Nonetheless, PAI-1 appears to play a role in obesity-related insulin resistance. 

Furthermore, PAI-1 has been shown to directly affect key nodes within the insulin signaling pathway. Balsara et al., [135] reported an increase in Akt^Ser473^ phosphorylation in PAI-1 deficient endothelial cells, isolated from mice aortic tissues, which could be attenuated in response to PAI-1 treatment. In agreement, Tamura et al. [48] showed a decrease in insulin-induced Akt^Ser473^ phosphorylation by PAI-1 in HepG2 cells, a liver hepatoma cell line. Furthermore, the authors also showed that downstream of Akt, PAI-1 increased the mRNA levels of two key gluconeogenic enzymes, G6Pase and PEPCK, suggesting PAI-1 could affect hepatic glucose metabolism [48]. 

Thus, in addition to being increased in response to T2D, insulin resistance, and obesity, evidence exists (Table 1) that PAI-1 may also contribute to the development of these conditions. 

### 4.2. Serum Amyloid A (SAA) 

SAA is a well-characterized APP that is predominantly synthesized in the liver [96,137]. It is an apolipoprotein that can bind and transport lipids in the blood and is mainly associated with high-density lipoproteins (HDLs) [137,138]. The important functional role of SAA during the APR, in host defense, has made it a sensitive marker of inflammation, in addition to CRP [102,139]. Indeed, during the APR, the plasma levels of SAA increase up to 1000-fold, from 1-5 µg/mL in healthy individuals, to exceeding 1 mg/mL in diseased patients [138,140]. Like PAI-1, SAA levels are increased in response to pro-inflammatory cytokines and GCs [30,31,32,35,36,141]. 

There are four different isoforms of the SAA gene (SAA1-4) of which SAA1 and SAA2 encode acute-phase SAA proteins and SAA4 is a constitutively expressed protein [138,140,142,143]. In humans, SAA3 is a pseudogene, but is functionally expressed in the adipose tissue of mice [143], particularly obese mice [144]. 

During the APR, SAA is secreted into circulation as a free protein and rapidly associates with HDLs, its physiological carrier [138]. The amphipathic structure of SAA facilitates its binding to HDLs and its ubiquitous diffusion via the circulation to all organs and tissues, to perform its biological function [96]. The association of SAA to HDLs during acute inflammation may also alter HDL metabolism and cholesterol transport [137,138,145]. The immune-related functions of SAA include acting as a chemoattractant for monocytes, leukocytes, and polymorphonuclear cells to inflammatory sites, resulting in the augmentation of inflammation [137,143,145]. These inflammatory functions of SAA are due to its ability to bind to various cell surface receptors [137,146], which results in the activation of various inflammatory signaling pathways, such as the MAPK pathways [146,147]. 

Like PAI-1, SAA, is a marker of obesity [148] and has been extensively studied with relation to this inflammatory condition (Table 2). Increased circulating levels of SAA have been observed in obese individuals, which positively correlates with an increased body mass index and decreased weight loss [148,149,150]. Additionally, like PAI-1, SAA has been shown to affect adipocyte differentiation in vitro by reducing the expression of adipogenic transcription factors [144,151]. SAA also induces the dysregulation of lipid metabolism, which is also associated with obesity, by increasing lipolysis [144,148,151] and decreasing lipid synthesis [151]. Mice fed a high-fat diet were protected from weight gain when treated with an anti-sense oligonucleotide that inhibits SAA mRNA expression, in addition to preventing adipose tissue expansion as well as macrophage infiltration into adipocytes [152]. Thus, not only are SAA levels increased in obesity, they also appear to play an active role in the development thereof. 

SAA is also a marker of T2D and insulin resistance [153]. Indeed, serum SAA concentrations of T2D patients are significantly increased, ranging from 2.1-24 µg/mL, which is comparable to levels observed in obese individuals [150,154,155,156]. Additionally, elevated plasma SAA levels (as well as other markers of inflammation including TNF-α, IL-6, and CRP) were observed in previously healthy individuals, who presented with onset T2D [43,44]. In diabetic mice, increased SAA mRNA levels correlate with chronic hyperglycemia [157]. Treatment of T2D patients with troglitazone not only inhibited hyperglycemia but also significantly reduced SAA levels [155]. These findings raise the question of whether SAA is more than just a biological marker for T2D or whether it could also contribute to its development. Scheja and colleagues investigated this hypothesis and found that in insulin resistance prone mice that were fed a high-fat diet, liver SAA1 and SAA2 mRNA levels, and adipose tissue SAA3 mRNA levels were increased. They also found that SAA decreased IRS-1 and GLUT-4 mRNA expression in 3T3-L1 adipocytes [153]. In accordance, others showed decreased IRS-1 tyrosine phosphorylation as well as decreased GLUT-4 protein expression and insulin-stimulated glucose uptake in 3T3-L1 adipocytes treated with SAA [144,158]. Taken together, these studies support the hypothesis that SAA may play a role in the development of insulin resistance, which could consequently lead to T2D. However, most of the studies investigated the effect of SAA in adipose tissue, and little research exists on how the liver or skeletal muscle is affected by SAA (Table 2). Additionally, the effect of SAA on other nodes of the insulin signaling pathway such as the IR and Akt is yet to be established.

### 4.3. C-Reactive Protein (CRP)

Discovered in 1930 in the serum of patients with acute pneumococcal pneumoniae [159], CRP was the first described APP. It was named for its capacity to bind the C polysaccharide of *Streptococcus pneumoniae* [100,139,160,161] and subsequently played a significant role in the identification of the APR [161]. CRP, also named pentraxin 1, is a member of the highly conserved pentraxin family of proteins, which include other structurally related molecules such as SAA. Like SAA, CRP is primarily synthesized by hepatocytes [162]. 

The main physiological role of CRP lies within the innate immune system, where it acts as an early defense system against foreign infectious pathogens. CRP exhibits anti-inflammatory activities including: (i) activation of the classical complement pathway, through binding to the C1q molecules, (ii) promoting apoptosis or phagocytosis of damaged cells and lastly (iii) displaying an anti-inflammatory effect by inhibiting neutrophil (leukocytes) action [162]. CRP participates in the systemic response to inflammation, increasing up to 1000-fold. Its levels start to rise after six to eight hours and peak by 48 h, after an inflammatory event [163]. CRP serum concentrations increase dramatically during acute and chronic inflammation, in response to a variety of inflammatory cytokines, including TNF-α and IL-6, and in some non-inflammatory conditions such as stress [164]. For this reason the measurement of CRP levels is widely used to monitor various inflammatory states [164]. Variable plasma levels, ranging from 0.8–3 µg/mL, are found in healthy individuals [162]. Factors such as polymorphisms in the CRP gene, could contribute to these variations [162]. However, CRP concentrations between 2 and 10 µg/mL are considered to indicate metabolic inflammation, which could lead to the development of insulin resistance [139]. This is supported by Festa and colleagues, who found a significant correlation between increased CRP levels and the development of T2D, with diabetic individuals having higher baseline levels of CRP (1.3–5.9 µg/mL) compared to the control group (0.8–3.4 µg/mL) [39].

Like PAI-1 and SAA, circulating levels of CRP have been studied in relation to insulin resistance and T2D, due to its role as a sensitive inflammatory marker. Several cross-sectional studies have shown that CRP levels are associated with obesity [165,166], increased fasting blood sugar levels [166], and impaired insulin sensitivity [167,168], all components of insulin resistance. These findings increased speculation that elevated CRP levels might be able to identify individuals in a prediabetic, insulin-resistant state [169]. In addition, several epidemiological studies have shown that increased CRP levels may predict the development of future T2D. For example, the Women’s Health Study (WHS) [42] demonstrated an association between CRP and insulin-resistant states, showing that among healthy women, high levels of IL-6 and CRP were associated with an increased risk for the development of T2D. In addition, the Cardiovascular Health Study (CHS) [41] also demonstrated that in a population of elderly men and women, elevated baseline CRP levels predicted the development of T2D. Finally, the IRAS, showed that high CRP baseline levels (>2.4 mg/L) amongst patients diagnosed with insulin resistance were associated with a higher risk of developing T2D [39] and recognized a significant correlation between CRP and components of insulin resistance [38]. 

In addition to establishing CRP as a predictive risk factor for insulin resistance and the development of T2D, numerous studies also investigated whether CRP could play a role in the development of the disease state (Table 3). Alessandris and colleagues demonstrated, using rat skeletal muscle cells, that high concentrations of CRP impaired insulin signaling by increasing IRS-1 serine phosphorylation and reducing the activation of Akt [50]. Additionally, this resulted in reduced glycogen synthesis and glucose uptake, thus, showing that CRP has an overall effect on the regulation of glucose metabolism. In agreement, Xu et al. showed a similar effect of CRP on insulin signaling in endothelial cells, reporting increased IRS-1 serine phosphorylation and decreased Akt activation [49]. Similarly, decreased IRS-1 tyrosine phosphorylation and its association with PI3K, as well as increased serine phosphorylation of IRS-1 in response to CRP was reported in primary rat hepatocytes as well as in vivo [51]. 

In summary, like the previously mentioned APPs, CRP is described as a strong predictor for the development of T2D [41,42,169,170]. Additionally, the role of CRP in the development of insulin resistance by affecting the insulin signaling pathway in hepatocytes, skeletal muscle, and endothelial cells has been described (Table 3) [49,50,51]. However, the effect of CRP on other key nodes in the insulin signaling pathway such as the IR have not been researched to fully elucidate its role in insulin resistance.

## 5. Regulation of the Acute Phase Proteins

The regulation of each APP is uniquely complex, with pro-inflammatory cytokines, GCs, and growth factors being some of its main mediators. Both in vitro and in vivo studies have reported the regulation of PAI-1, SAA, and CRP expression to be closely influenced by the pro-inflammatory cytokines, TNF-α, IL-1β and IL-6, as well as hormones such as GCs which are also associated with insulin resistance and T2D [107,171,172,173,174,175,176,177,178,179]. For example, patients diagnosed with Cushing’s syndrome, which is associated with GC excess, often also present with insulin resistance and T2D [180]. GCs impair insulin signaling, and long-term exposure also negatively affects pancreatic beta-cells from secreting insulin [28]. Likewise, low-grade chronic inflammation associated with obesity and the subsequent increase in pro-inflammatory cytokine secretion is associated with insulin resistance [181], with TNF-α, IL-1β, and IL-6 directly impairing insulin signal transduction [16,23,24,26,27].

IL-6 and IL-1 have been reported to enhance PAI-1 transcription in hepatoma cell lines and whereas IL-6 induced a modest increase in PAI-1 mRNA levels, IL-6 in combination with IL-1 had a much greater effect on PAI-1 mRNA expression [182,183]. TNF-α enhanced PAI-1 mRNA and protein expression in endothelial cells [184,185], but seems to affect PAI-1 mostly in adipose tissue, both in vitro and in vivo [186,187,188] by increasing mRNA levels [186] as well as PAI-1 activity and protein expression [187,188]. Interestingly, TNF-α-induced PAI-1 protein expression is enhanced in combination with insulin [188], which also stimulates PAI-1 transcription and protein synthesis in a number of different cell models [185,188,189,190]. These studies suggest that TNF-α (which is related to obesity) might be the key inducer of PAI-1 expression in adipose tissue in obesity-related insulin resistance. 

As markers of inflammation and major positive APPs, SAA and CRP expression are mainly regulated by IL-6, IL-1β, and TNF-α. However, several in vitro studies investigating SAA and CRP mRNA and protein expression in hepatoma cell lines, show differential regulation by these cytokines. For instance, SAA mRNA expression is induced by all three cytokines, however to different extents [30,31,32,36,191,192,193,194]. IL-1β was shown to be a strong inducer of SAA mRNA expression [36,193], whilst IL-6 and TNF-α stimulates SAA mRNA expression to a lesser extent [32,194,195]. TNF-α and IL-6 in combination, however, enhanced SAA mRNA expression [32]. Furthermore, TNF-α, IL-1β, and IL-6 in combination were able to enhance the transcription of SAA to a greater extent compared to any single treatments [195]. CRP synthesis, on the other hand, was shown to be mainly regulated by IL-6 in the hepatoma cell lines [33,192]. In primary human hepatocytes, IL-1β was able to upregulate CRP synthesis, via inducing the synthesis of IL-6, strengthening the argument that CRP levels are mainly upregulated by IL-6 in the liver [33]. Interestingly, TNF-α alone, or in combination with IL-6, had no effect on CRP synthesis [30].

The induction of SAA and CRP is not limited to the liver. CRP production was induced by IL-1 and IL-6, alone, and in combination in human adipocytes [196], whereas SAA3 mRNA expression was increased in response to IL-1β, TNF-α, and IL-6 in 3T3-L1 adipocytes [34,197]. It was found that the positive effect on SAA3 mRNA expression induced by IL-6 and IL-1β was mediated by JNK and NFκB, respectively [34,197] two proteins which negatively regulate insulin signaling [198,199]. 

The anti-inflammatory GCs also regulate PAI-1, SAA, and CRP expression [37,48,200,201,202]. Several studies have shown an increase in PAI-1 mRNA and protein expression in response to the synthetic GC, dexamethasone [37,201,202,203]. Interestingly, dexamethasone potentiates TNF-α-induced PAI-1 mRNA expression in epithelial cells [37]. However, it is not yet known whether this combinatorial effect is cell specific or if dexamethasone can enhance IL-6 or IL-1β-induced PAI-1 expression. Furthermore, corticosterone, the endogenous GC in rodents, increased both PAI-1 mRNA and protein levels in vivo [48]. 

Like PAI-1, the cytokine-driven production of SAA and CRP in hepatoma cell-lines can be potentiated by GCs [31,32,33,34,35,191,204,205]. Dexamethasone treatment in combination with TNF-α, IL-1β, or IL-6 increased SAA and CRP production to a greater extent in comparison to the respective cytokine alone [30,31,32,33,35,141,206]. 

The fact that the levels of these APPs are induced by both pro-inflammatory cytokines and GCs is interesting considering that GCs are mostly known for their anti-inflammatory properties [207]. Traditionally GCs and the majority of pro-inflammatory cytokines antagonize each other’s activity [208]. However, current knowledge suggests that GCs selectively regulate gene expression [204]. When it comes to innate immune responses such as the APR, GCs display pro-inflammatory behavior, converging their signal with that of pro-inflammatory cytokine signaling, to further increase the expression of certain APPs. Ultimately, by doing so, GCs reinforce the innate immune system and the APR [209]. 

## 6. Conclusions

Numerous factors contribute to the development of insulin resistance and subsequently T2D, such as obesity and stress, with inflammation a key role player. APPs, which are markers of inflammation, have been closely associated with T2D as their serum levels are elevated in T2D patients [44,92,150,154,155,156]. These include PAI-1, SAA, and CRP, which all play different roles in response to inflammation such as opsonization, activating the complement system modulating the host’s immune response, and aiding in repairing damaged tissue thereby establishing homeostasis during the APR [210].

Whether these APPs are just biological markers for T2D or actually influence the development of insulin resistance (and are not just correlative) is still unclear. Some studies support the possibility that PAI-1, SAA, and CRP impair insulin signaling directly [45,46,47,48,49,50,51,135,144,153,158], whilst others believe that APPs are only correlated with T2D [54,111,211,212,213]. 

As the levels of these APPs are also regulated by pro-inflammatory cytokines and GCs, both of which are also associated with T2D development [16,29], we speculate that APPs may be the causative link between the physiological risk factors (stress and inflammation) and the development of insulin resistance (Figure 1). Thus, APPs could contribute to the manifestation of pro-inflammatory cytokine and GC-induced insulin resistance, adding to the complexity of inflammatory- and GC-induced insulin resistance. This also suggests a cumulative effect of stress- and inflammatory mediators together with circulating APPs to induce insulin resistance. Therefore, understanding the role of these APPs in insulin resistance and T2D progression could provide insight into novel mechanisms of action that lead to the development of insulin resistance and towards the development of innovative drug targets.

## Figures and Tables

**Figure 1 cells-11-02163-f001:**
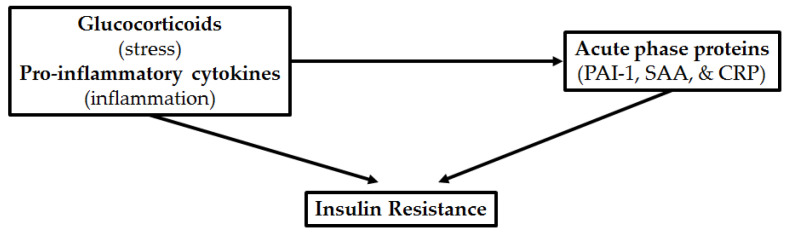
Acute phase protein, plasminogen activator inhibitor-1 (PAI-1), serum amyloid A (SAA), and C-reactive protein (CRP), expression is regulated by glucocorticoids and pro-inflammatory cytokines. APPs may be the causative link between stress and inflammation and the development of insulin resistance.

**Table 1 cells-11-02163-t001:** Studies supporting the role of PAI-1 in the development of obesity, insulin resistance, and type-2 diabetes.

Disease State	Model System	Supporting Data	Reference
**Obesity**	In vivo Primary cultured adipocytes from PAI-1-deficient (PAI−/−) mice and overexpressed (PAI+/+) mice	PAI-1 deficiency: Enhanced adipocyte differentiation Enhanced insulin-stimulated glucose uptake PAI-1 overexpression: Adipocyte differentiation inhibited Reduced PPARγ expression.	Liang et al., 2006 [132]
	In vivo High-fat diet-induced obesity in PAI-1 knockout mice	PAI-1 deficiency: Fat accumulation prevented PPARγ expression in adipocytes maintained	Ma et al., 2004 [46]
	In vivo Diet-induced obesity in mice, administered the PAI-1 inhibitor, PAI-039 In vitro Human pre-adipocytes treated with the PAI-1 inhibitor, PAI-039	PAI-1 inhibition: Dietary fat-induced obesity attenuated Lower glycemia and triglyceride level showed PAI-1 inhibition: Human pre-adipocyte differentiation attenuated	Crandall et al., 2006 [130]
	In vivo Genetic model of obesity and diabetic mice lacking the PAI-1 gene	PAI-1 deficiency: Murine adiposity reduced	Schäfer et al., 2001 [45]
	In vivo Diet-induced obesity in PAI-1 deficient mice	PAI-1 deficiency: Faster weight gain in PAI-1 deficient mice	Morange et al., 2000 [134]
	In vivo Transgenic mice with overexpression of PAI-1 in adipose tissue, administered the PAI-1 inhibitor, PAI-039	PAI-1 overexpression: Adipose tissue growth impaired PAI-1 inhibition: Adipose tissue development unaffected Improved insulin sensitivity in wildtype mice	Lijnen et al., 2005 [131]
**Insulin Resistance**	In vivo PAI-1 knockout mice fed a high-fat diet	PAI-1 deficiency: Decreased the plasma glucose, insulin and cholesterol levels that were markedly increased by the high-fat diet	Tamura et al., 2014 [47]
	In vitro HepG2 cells were treated with 20 nM PAI-1 for 24 h	PAI-1 treatment: Hepatic insulin signaling affected Decreased insulin-induced glucose uptake Gluconeogenesis affected through the increase of G6Pase and PEPCK mRNA levels	Tamura et al., 2015 [48]
	In vitro PAI-1 knockout endothelial cells treated with 10 ng/mL PAI-1 for 24 h	PAI-1 deficiency: Increased Akt activation PAI-1 treatment: Decreased Akt activation	Balsara et al., 2006 [135]
	In vivo High-fat diet-induced obesity in PAI-1 knockout mice	PAI-1 deficiency: Glucose uptake increased Plasma glucose and insulin levels maintained	Ma et al., 2004 [46]
	In vivo Genetic model of obesity and diabetic mice lacking the PAI-1 gene	PAI-1 deficiency: Hyperglycemia and hyperinsulinemia associated with insulin resistance improvement	Schafer et al., 2001 [45]
	In vitro 3T3 adipocytes treated with 100nM PAI-1 in the presence of insulin and vitronectin	PAI-1 treatment: Decreased Akt activation	López-Alemany et al., 2003 [136]
**T2D**	Epidemiological study The IRAS—measured PAI-1 levels in non-diabetic patients in relation to incident diabetes within 5 years	Elevated levels of PAI-1 (±24 ng/mL) were associated with incident T2D.	Festa et al., 2002 [39]
	Epidemiological study Follow up study to Festa et al. 2002.	Progression of PAI-1 levels over time, in addition to high baseline levels (23.7 ng/mL), was associated with the onset of T2D	Festa et al., 2006 [40]

**Table 2 cells-11-02163-t002:** Studies supporting the role of SAA in the development of obesity, insulin resistance, and type-2 diabetes.

Disease State	Model System	Supporting Data	Reference
**Obesity**	In vitro 3T3-L1 adipocytes	SAA treatment: Decreased adipocyte differentiation: by decreasing adipogenic transcription factors (PPARγ, C/EBPα) Increased lipolysis	Filipin-Monteiro et al., 2012 [144]
	In vivo SAA mRNA inhibition in mice fed a high-fat diet	SAA inhibition: Adipose tissue expansion inhibited Macrophage infiltration into adipose tissue inhibited	De Oliveira et al., 2016 [152]
	In vivo Serum SAA levels in obese individuals In vitro Human adipocytes treated with SAA (2.34 µg/mL) for 24 h	SAA levels increased in obese individuals. SAA levels decreased after weight loss. SAA treatment: Increased lipolysis	Yang et al., 2006 [148]
	In vitro Human adipocytes treated with SAA for 24 h	SAA treatment: Increased lipolysis Reduced mRNA expression of transcription factors (PPARγ and C/EBPα) involved in adipocyte differentiation Reduced mRNA expression of SREPB-1c which is involved in lipid synthesis	Faty et al., 2012 [151]
	In vivo Gene expression in obese individuals	Increased expression of SAA1 and SAA2 mRNA and protein expression in obese individuals.	Poitou et al., 2005 [149]
**Insulin resistance**	In vitro 3T3-L1 adipocytes	SAA treatment: Insulin-stimulated glucose uptake decreased	Filipin-Monteiro et al., 2012 [144]
	In vitro 3T3-L1 adipocytes	SAA treatment: Decreased mRNA expression of Glut4 and IRS-1	Scheja et al., 2008 [153]
	In vitro 3T3-L1 adipocytes	SAA treatment: Reduced insulin-stimulated glucose uptake Decreased IRS-1 activation Decreased GLUT4 expression	Ye et al. 2009 [158]
	In vivo SAA mRNA inhibition in mice fed a high-fat diet	SAA inhibition: Protected mice from weight gain and insulin resistance.	De Oliveira et al., 2016 [152]
**T2D**	In vivo Diabetic (ob/ob) mice. Measured SAA3 mRNA in adipose tissue	Isolated adipose tissue of T2D mice showed drastically increased SAA3 mRNA levels.	Lin et al., 2001 [157]
	Epidemiological study Patients with T2D who received daily treatment with troglitazone (anti-diabetic drug)	SAA levels were above the range for healthy subjects (approx. 6.2 µg/mL). Troglitazone reduced SAA levels (by 25% down to 4.0 µg/mL).	Ebeling et al., 1999 [155]
	Epidemiological study Measured SAA levels in patients with individuals with impaired glucose tolerance in comparison with individuals with and without T2D	Plasma levels of SAA were significantly higher in patients with T2D and impaired glucose tolerance (approx. 6 µg/mL).	Müller et al., 2002 [43]
	Epidemiological study Measured SAA levels in non-diabetic individuals who participated in a 7-year follow-up	SAA levels were significantly associated with the onset of T2D (approx. 4.0 µg/mL).	Marzi et al., 2013 [44]
	Epidemiological study Measured SAA levels in T2D patients	Insulin resistance and T2D was significantly correlated with SAA levels (approx. 24 µg/mL).	Leinonen et al., 2003 [156]

**Table 3 cells-11-02163-t003:** Studies supporting the role of CRP in the development of insulin resistance and type-2 diabetes.

Disease State	Model System	Supporting Data	Reference
**Insulin resistance**	In vitro Rat skeletal muscle (L6) cells treated with 10 mg/l CRP	CRP treatment induced insulin resistance in skeletal muscle cells by: Increasing serine phosphorylation of IRS-1 Reducing activation of Akt Reducing glycogen synthesis Impairing glucose uptake	Alessandris et al., 2007 [50]
	In vitro Mouse endothelial cells treated with recombinant CRP at various doses and times	Overall CRP impaired insulin signaling in endothelial cells by: Increasing serine phosphorylation of IRS-1 Decreasing activation of Akt	Xu et al., 2007 [49]
	In vitro Primary cultured rat hepatocytes treated with 30 mg/L CRP In vivo Rats treated with CRP	CRP induced hepatic insulin resistance both in vivo and in vitro by: Reducing the activation of IRS-1 and Akt Impairing the association of IRS-1 with PI3K Inducing the inhibition of IRS-1 (through serine phosphorylation)	Xi et al., 2011 [51]
**Type-II diabetes**	Epidemiological study The IRAS study—measured CRP levels in non-diabetic patients in relation to incident diabetes within 5 years	Elevated CRP levels (>2.4 mg/L) was associated with incident T2D.	Festa et al., 2002 [39]
	Epidemiological study Measured insulin sensitivity and CRP levels in the non-diabetic population of the IRAS study	Elevated CRP levels (>3.53 mg/L) was strongly associated with components of insulin resistance and T2D.	Festa et al., 2000 [38]
	Epidemiological study Women’s Health Study	High CRP levels were associated with increased risk for development of T2D.	Pradhan et al., 2001 [42]
	Epidemiological study Cardiovascular Health Study	High baseline levels (2.8 mg/L) of CRP predicted T2D.	Barzilay et al., 2001 [41]

## Data Availability

Not applicable.

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
