# Peer review of "The Association of Acute Phase Proteins in Stress and Inflammation-Induced T2D"

_cells, 2022, doi:10.3390/cells11142163_

Round 1
Reviewer 1 Report
In this review the authors proposed that, rather than correlative, the Acute Phase Proteins (APPs), PAI-1, SAA, and CRP, could play a causative role in the development of obesity-associated inflammation, insulin resistance and finally type 2 diabetes. Although the authors stated along the manuscript that more work must be done to support the causative link between PAI-1, SAA, and CRP and the development of insulin resistance and inflammation, I only found a few references supporting that APPs do not play a causative role in the development of obesity-associated inflammation, insulin resistance and type 2 diabetes. Hence, I would suggest including some more recent references regarding this aspect (doi: 10.1371/journal.pone.0266688).
On the other hand, in p.4, lines 156-159 according to data in reference 43, the authors stated that PAI-1 levels are markedly increased parallel with fat accretion, explaining the correlation between elevated circulating PAI-1 levels and obesity, a risk factor for type 2 diabetes. Nevertheless, the referenced work was reported in 2014, but previous work reported in 2009 demonstrated the association between visceral adiposity accretion and plasma PAI-1 levels in old insulin-resistant rats (doi:10.1152/ajpregu.00093.2009) and this should be considered.
Finally, in Table 1 there is some missing information about the Model system and the Supporting data regarding the reference (123) López-Alemany et al 2003.
Author Response
Dear Reviewer,
Thank you for your careful consideration and review of our manuscript entitled "The Association of Acute Phase Proteins in Stress and Inflammation-induced T2D"
Please see below our responses to each of your comments:
- We have added your suggested reference Ji et al 2022 in addition to Ahlin et al 2013 (Line 430 of manuscript), which demonstrated that adipose tissue derived SAA had no effect on the development of insulin resistance.
- We have also added your suggested reference Serrano R et al 2019 with the addition of two earlier studies Ma et al 2004, and Shafer et al 2001, which have also shown a link between PAI-1 and adiposity and insulin sensitivity to the already included Tamura et al 2014. (Line 167 of manuscript)
-
Apologies for this oversight. The relevant information was added to the table.
In vitro
3T3 adipocytes treated with 100nM PAI-1 in the presence of insulin and vitronectin
PAI-1 treatment:
- Decreased Akt activation
López-Alemany et al. 2003 [126]
Reviewer 2 Report
The manuscript is about “The Association of Acute Phase Proteins in Stress and Inflammation-induced T2D”.
Comments needs to be addressed before accepted for publication:
Provide the justification of Acute phase proteins and its importance. Authors must provide adequate details of Insulin Resistance with recent citations. Overall, the manuscript is well written with adequate details.
Conclusion of the manuscript is not informative so it has to be revised.
Author Response
Dear Reviewer,
Thank you for your careful consideration and review of our manuscript entitled "The Association of Acute Phase Proteins in Stress and Inflammation-induced T2D"
Please see below our responses to each of your comments:
- Provide the justification of Acute phase proteins and its importance.
-
The primary biological activity of each acute phase protein relevant to this manuscript is discussed in this manuscript under their various sections i.e 4.1 (PAI‑1 (lines 169-179)), 4.2 (SAA (lines 245-254)), and 4.3 (CRP (lines 299-310)) due to each APP possessing unique physiological activities. They all contribute to restoring homeostasis upon initiation of the acute phase response, which is discussed in section 3.
This is highlighted in the manuscript (lines 149-152 of manuscript) in the following statement:
“The biological functions of the different positive APPs are vast and involve activating the complement system (which also plays a role in T2D progression), modulating the host’s immune response as well as wound healing and tissue repair.”
Nonetheless, to further highlight the relevance of acute phase proteins the following sentence was adapted “Overall the APR is important to restore homeostasis and lack of resolution of the inflammatory stimulus results in chronic inflammation.” To the following:
“Overall, the APR involving various APPs (each with a unique set of biological activities) is important to restore homeostasis and lack of resolution of the inflammatory stimulus results in chronic inflammation.” (Line 153-155 in the manuscript)
Additionally, in the conclusion, we have expanded the following sentence “These include, PAI-1, SAA, and CRP, which all play different roles during the APR.”
To state the following: “These include PAI-1, SAA, and CRP, which all play different roles in response to inflammation such as opsonization, activating the complement system modulating the host’s immune response, and aiding in repairing damaged tissue thereby establishing homeostasis during the APR.” (Line 422-425 in the manuscript)
-
- Authors must provide adequate details of Insulin Resistance with recent citations.
-
Insulin resistance is discussed in section 2 of the manuscript (lines 69-125 of the manuscript). In this section we mention the characteristics of insulin resistance including the proposed molecular mechanism of action involving various mediators within the insulin signalling pathway. We also highlight how the liver, adipose tissue and skeletal muscle respond differently in response to insulin insensitivity.
We have added these recent references Lee et al 2022 (doi 10.2337/DB18-0856), Freeman & Pennings 2021 (https://www.ncbi.nlm.nih.gov/books/NBK507839/) and James et al 2021 (doi 10.1038/s41580-021-00390-6) to the following sentence. “The characteristic attenuated effect of insulin in peripheral tissues, indicative of insulin resistance, precedes the development of hyperglycemia.”(lines 69-70 in the manuscript)
Also, after the following sentence : “Defective insulin action manifests itself as reduced glucose uptake in skeletal muscle and adipose tissue and increased glucose production in the liver, amongst other out-comes.” we added the following sentence (lines 72-76 of manuscript)
“More specifically, insulin-induced glucose uptake via the glucose transporter type 4 (GLUT4) is restricted in both skeletal and adipocytes in an insulin-resistant state [Leto et al 2012]. Additionally, glycogen synthesis in response to insulin is no longer promoted in the insulin-resistant liver and skeletal tissue and glycogenolysis is not suppressed [Petersen & Shulman 2018].”
Lastly, a more recent reference Mezza et al 2019 (doi 10.2337/DB18-0856) was added to the following sentence “Finally, when the β-cells, due to β-cell dysfunction, fail to produce the excess amounts of insulin needed, T2D emerges” (lines 81-82 of manuscript).
-
- Conclusion of the manuscript is not informative so it has to be revised.
-
The conclusion of the manuscript brings together the different concepts mentioned in the previous section. Highlighting our hypothesis that “APPs may be the causative link between the physiological risk factors (stress and inflammation) and the development of insulin resistance (lines 439-440 in the manuscript)”, which is also illustrated in figure 1. We propose that APPs contribute to inflammatory- and glucocorticoid-induced insulin resistance and that this should be considered in future investigations.
-
To emphasize the implication of our hypothesis we have added the following “adding even more to the complexity of inflammatory-and GC-induced insulin resistance.” (Line 442 in the manuscript) to the sentence “Thus, APPs could contribute to the manifestation of pro-inflammatory cytokine and GC-induced insulin resistance”
Additionally, following this we added the following sentence “This also suggests a cumulative effect of stress- and inflammatory mediators together with circulating APPs to induce insulin resistance.” (Lines 442-444 in the manuscript)
-
Reviewer 3 Report
This manuscript is a review article, aimed to analyse weather acute phase proteins (APPs), such as plasminogen activator inhibitor-1 (PAI-1), serum amyloid A (SAA), and C-reactive protein (CRP), are causative and nor correlative link between the physiological elements of risk (stress and inflammation) and the development of insulin resistance, which has leading role in pathogenesis of T2D. In that sense, the article has clinical importance for everyday clinical practice and summarise important up to date data. In order to clarify this issue, the authors have done the complicated work of critical review of published studies. However, there are some issues to be clarified.
1. Line 23….do not start the sentence with abbreviation …..
2. Line 23 …DM is classified as either: i) gestational DM, ii) type-I DM (T1D) or 23 iii) type- II DM (T2D)…..it is not correct Roman numbers are not in use any more, just type 1 and type 2 diabetes
3. Line 25… the sentence… Although current treatments for T2D are often effective, these treatments are linked to gastrointestinal side-effects and increased cardiovascular complications [3,4]…. should be reformulated …. The current therapy of T2D might be ineffective, but it does not induce complications per se, and someone could conclude that from this sentence…
4. You must add the key reference about insulin resistance …Reaven GM. Banting lecture 1988. Role of insulin resistance in human disease. Diabetes. 1988 Dec;37(12):1595-607. doi: 10.2337/diab.37.12.1595. PMID: 3056758.
Author Response
Dear Reviewer,
Thank you for your careful consideration and review of our manuscript entitled "The Association of Acute Phase Proteins in Stress and Inflammation-induced T2D"
Please see below our responses to each of your comments:
- Line 23….do not start the sentence with abbreviation …..
-
DM was changed to Diabetes mellitus (line 22 of manuscript).
-
- Line 23 …DM is classified as either: i) gestational DM, ii) type-I DM (T1D) or 23 iii) type- II DM (T2D)…..it is not correct Roman numbers are not in use any more, just type 1 and type 2 diabetes
-
Thank you for this suggestion as we were unaware of this. We have replaced all roman numerals with the more contemporary usage. (line 23 of manuscript)
-
- Line 25… the sentence… Although current treatments for T2D are often effective, these treatments are linked to gastrointestinal side-effects and increased cardiovascular complications [3,4]…. should be reformulated …. The current therapy of T2D might be ineffective, but it does not induce complications per se, and someone could conclude that from this sentence…
-
Various side-effects are associated with current T2D treatments (reviewed in Padhi S et al 2020; doi 10.1016/J.BIOPHA.2020.110708) which deserves to be mentioned. However, taking into consideration your comment we have addressed any alarmist rhetoric by the following changes where we highlighted specifically which side-effect is associated with which treatment.
“Although current treatments for T2D are often effective, they are linked to various side effects. For example, Metformin, a biguanide, commonly described in patients diagnosed with T2D is linked to gastrointestinal side effects. The usage of Rosiglitazone, once widely prescribed to treat T2D, is currently restricted in most countries due to cardiovascular complications. Therefore, novel therapeutic approaches are warranted.” (lines 25-31 in the manuscript).
-
- You must add the key reference about insulin resistance …Reaven GM. Banting lecture 1988. Role of insulin resistance in human disease. Diabetes. 1988 Dec;37(12):1595-607. doi: 10.2337/diab.37.12.1595. PMID: 3056758.
-
This important reference has been added to the following sentence (lines 32-33 in the manuscript):
“T2D, a major non-communicable disease, is traditionally considered a metabolic disorder, which is mainly attributed to the initial development of insulin resistance”
Additionally, we also added this reference to the following sentence (lines 36-37 in the manuscript):
“Although it is well established that insulin resistance is central to the pathogenesis of T2D.”
-